# FILI: SYNTAX REPAIR BY LEARNING FROM OWN MISTAKES

## ABSTRACT

Automatically fixing syntax errors in programs is a key challenge in Software Engineering community. Although, there are millions of programs on the web, both syntactically correct and incorrect, finding a large number of paired examples of ⟨correct, incorrect⟩ programs is difficult. This makes training a program fixer using supervised learning difficult. Recently, BIFI, an unsupervised approach for learning a syntax fixer was proposed, in which an additional model (Breaker model) is used to augment data in each learning iteration to match real-world error distribution. In this paper, we propose a novel approach, FILI (Fix-It-Learn-It) for learning a syntax fixer without having to train any additional models for data augmentation. In each iteration, FILI carefully selects examples from the fixer's own predictions, both correct and incorrect, and uses those to fine-tune the fixer. We also show that gradually increasing the complexity of the examples during training leads to a more accurate fixer. Our evaluation on the Github-Python dataset shows that FILI outperforms BIFI by **1%** while being significantly easier to train. Moreover, FILI avoids training the breaker model in each iteration, which can take about 2 days on a modest DNN accelerator.

## 1 INTRODUCTION

Automated program repair has long been a challenging problem in software development (Goues et al., 2021). One particular class of problem in program repair is the task of fixing syntax errors. A syntax error in a program occurs when a user attempts to compile a program that does not conform to the grammar of the programming language. When a syntax error occurs, the compiler halts the compilation and throws an error message, which may include the line number and offset of the error depending on the language. Often, these error messages are not very informative, and they may also point to a location other than the one where the error occurred, making these errors difficult to fix. This whole cycle of finding errors, fixing them, and re-compiling has a negative impact on programmer's productivity, especially for beginner programmers.

One of the simplest approaches to fixing syntax errors is to define rules for each class of errors and use them to automatically fix the errors. However, this rule-based approach is challenging because it neccessitates comprehensive knowledge of programming language's grammar and, often multiple possible fixes exists for the same incorrect program. As a result, manually writing rules for each error case becomes impractical. To address this, several approaches, both symbolic (constraint-based) (Singh et al., 2013) and learning-based (Bhatia et al., 2018; Gupta et al., 2017; Yasunaga and Liang, 2021; 2020; Pu et al., 2016), have been proposed for automatically fixing the syntax errors in a program. Learning-based approaches have shown promise, as they leverage data to learn patterns and automatically generate fixes. This approach offers several advantages, such as suggesting likely fixes based on prior examples and not requiring explicit domain knowledge of the programming language.

Learning-based approaches formulate the syntax fixing problem as machine translation problem to translate an incorrect program to a correct one, and various encoder-decoder architectures have been used in supervised settings to perform this translation. Because of the unavailability of large number of paired examples ⟨incorrect program, correct program⟩ for supervised training, recently, *Break-It-Fix-It* (BIFI), an unsupervised learning algorithm (Yasunaga and Liang, 2021) was proposed to overcome the lack of quality paired examples. BIFI uses two trained models: *a) Fixer* - which attempts to generate a syntactically correct program given an incorrect program as input, and *b) Breaker* - which

attempts to generate a syntactically incorrect program given a correct program as input. Starting from a fixer trained on synthetic data and real-world unpaired good (syntactically correct) and bad (syntactically incorrect) programs, BIFI improves the fixer by performing the following four steps in each iteration: *1)* Applies the fixer on the bad programs ($B$) to generate the corresponding good programs ($G'$), *2)* Trains the breaker using $\langle G', B \rangle$, *3)* Applies the breaker on good programs ($G$) to generate the corresponding bad programs ($B'$), and *4)* Trains the fixer with data $\langle B, G' \rangle$ from step 1 and $\langle B', G \rangle$ from step 3. This cycle of self-learning and data augmentation leads to the improvement of fixer's ability to solve previously unseen problems. Moreover, the breaker model, also learns to better match the distribution of real-world syntax errors in each iteration.

While BIFI requires training an additional breaker model in each iteration for data augmentation, in this paper, we propose *Fix-it-Learn-it* (FILI), a new self-learning approach in the context of syntax fixing. FILI is inspired by how programmers learn to fix errors in the real world. Programmers typically improve their skills by recognizing their own mistakes, gaining insights from them, and developing a deeper understanding of the programming language. Gradually, they accumulate knowledge, enabling them to identify and fix more complex errors with greater accuracy and confidence.

Similar to BIFI, FILI improves the fixer with each iteration by training on examples it can already fix. In addition, FILI improves the fixer by learning from its *own* mistakes. These examples are generated using beam search which maintains a set of the most likely hypotheses at each step of the decoding process. We identify the fixer's predictions from the beam that do not parse and pair them with the programs from the beam that are fixed. FILI starts from a fixer trained on synthetic data and in each iteration performs the following steps *1)* Applies the fixer on the bad programs ($B$) to generate a beam consisting of most likely predictions, *2)* Identifies using a parser the good programs ($G$) and the bad programs ($B'$) from the beam, *3)* Trains the fixer on the paired data $\langle B, G \rangle$ and $\langle B', G \rangle$.

In contrast to BIFI, FILI does not require training an additional *breaker* model to augment data at each iteration. We hypothesize that it is not necessary to precisely match the distribution of the real-world errors as long as the *fixer* improves its ability to fix different classes of errors with each iteration. While BIFI explored using a separate *breaker* model to augment data to improve its *fixer*'s performance, in this paper, we propose sampling examples from the *fixer*'s beam predictions which empirically turns out be simpler and more efficient than training a separate breaker model. We believe that this is effective because these negative programs are akin to sampling data from the decision boundary of the model. Training the *fixer* with these programs improves its confidence in handling these errors, effectively pushing them further down in the beam predictions.

In addition to learning from its own mistakes, FILI also adopts the curriculum learning style for fixing errors from real-world programmers. During training, we gradually increase the complexity of the examples used to train the fixer. The complexity of the examples is defined by the Levenshtein edit-distance between the bad and good programs. We begin by training with smaller edit-distances and gradually add examples with larger edit-distances. This eases the learning process as the fixer *a)* learns to fix errors incrementally, mimicking the cycle of identifying, fixing, and re-compiling, and *b)* generates programs by making minimum number of changes to the bad code.

We evaluate FILI on the open-source **GitHub-Python** dataset (Yasunaga and Liang, 2021). Our approach improves the accuracy of the fixer by $\sim 4\%$ when compared to the fixer trained using self-learning alone and by $\sim 1\%$ when compared to the state-of-the-art fixer that is trained using a breaker. A key contribution of our work is to significantly simplify the process of training a syntax fixer of (slightly) higher quality than prior work (viz., BIFI).

In summary, this paper makes the following contributions:

- We present a new approach in Section 4.2, FILI, for learning a *fixer* for syntax error correction in an unsupervised setting by augmenting examples from the *fixer*'s own prediction where it makes a mistake.

- We develop a curriculum in Section 4.3 i.e, by starting out with simpler examples (fewer program edits) and gradually increasing the complexity (more program edits) that results in a *fixer* which is more accurate in fixing errors .

- We evaluate FILI on real-world syntax correction tasks in Section 5, and show that while being simpler and computationally more efficient to train than previous approaches such as BIFI, it still outperforms them.

## 2    RELATED WORK

Automated Program Repair (Goues et al., 2021) is the task of automatically repairing an incorrect program given some correctness specification and is an active research area where several different techniques have been proposed ranging from constraint-based (Singh et al., 2013), genetic programming (Le Goues et al., 2012), and learning-based (Bhatia et al., 2018; Gupta et al., 2017; Yasunaga and Liang, 2021; 2020; Pu et al., 2016). These approaches aim to tackle different class of program errors such as syntax errors, semantic errors, logical errors, runtime errors, race conditions, etc.

In this paper, we focus on syntax errors only. BIFI (Yasunaga and Liang, 2021) is the closest work to ours and is also an unsupervised self-learning approach. The key difference between FILI and BIFI is that FILI does not require training a separate *breaker* model and iteratively learns from its own mistakes. SynFix (Bhatia et al., 2018) present an approach to train an RNN-based language model over a corpus of syntactically correct programs and uses the language model to generate potential corrections for errors identified by a parser. DeepFix (Gupta et al., 2017) trains an attention-based encoder-decoder model, where the encoder encodes an incorrect program token sequence, and the decoder generates the correction as a line number together with the fixed line. Unlike SynFix and DeepFix, FILI uses self-supervised learning to iteratively improve the fixer performance.

The use of negative samples from beam predictions to supplement training data has been explored recently by (Cao et al., 2021) for the grammatical error correction problem. Their approach pairs source sentences with beam predictions that are dissimilar to the target sentence, creating negative pairs alongside the ground-truth sentence pairs. In contrast, our approach differs in two key ways: 1) we do not rely on ground-truth sentence pairs to generate additional data from model's predictions, and 2) we do not use an additional contrastive loss to train with these additional examples.

Recently, there has been significant advancements in utilizing large language models for code-related tasks. These models with billions of parameters require massive compute for fine-tuning on new datasets. One challenge with these models is their tendency to hallucinate outputs if they are not confident about the given task. Some of the recent works (Chen et al., 2023; Shinn et al., 2023) have explored approaches to teach these models to self-debug, enabling them to identify their own mistakes and fix it. These approaches involve explaining the generated text in natural language or teaching the models to self-reflect when hallucinations are detected. These approaches share a similar goal of learning to correct their own mistakes, which is similar to our approach in this work. However, we focus on smaller models that are more accessible, as they can be trained and deployed on commodity hardware.

In unsupervised learning, pseudo-labelling (Lee et al., 2013), is used for augmenting training data. It involves initially training a model using labelled data and subsequently utilizing the trained model to label the unlabelled data based on probabilities. In contrast, in FILI we can assign real labels to unlabeled data using the compiler.

## 3    PROBLEM FORMULATION

In this section, we provide an overview of the problem formulation. Given a set $X$ of programs and a compiler $\mathcal{C}$ to check whether or not the program parses i.e., throws a syntax error or not. We use $X^+$ to represent the set of programs that parse (good programs) and $X^-$ the set of programs that have syntax errors (bad programs). The compiler is represented as $\mathcal{C} \colon X \rightarrow \{0, 1\}$, where the indicator function $\mathcal{C}\{x\}$ of program $x$ maps to 1 if the program compiles and to 0 if it throws an error. Our goal is to train a *fixer* $F$ which takes a bad program as input and generates the corresponding good program. This can be expressed as $F(x^+ \mid x^-)$ which represents the conditional probability of the fixer generating the good program $x^+$ given the bad program $x^-$. $F$ is a probabilistic model and we sample programs using beam search from this model. It should be noted that we do not have access to paired $\langle X^-, X^+ \rangle$ for training the *fixer* in supervised setting. Instead, we have access to a large collection of unpaired bad and good programs.

The *fixer* can potentially correct a program by deleting the entire line that contains the error or by deleting the entire program. We use the Levenshtein edit-distance $\delta$ metric between the bad and good programs to ensure that the model does not learn to make arbitrary large changes to the program. Given a program $x$, the sequence $\langle x_1, ..., x_n \rangle$ represents the tokenized program. We compute the

edit-distance $\delta$ at the program's token level. For instance, given two simple expressions $c = a + b$ and $c = var + b$, the edit-distance $\delta$ at the program string level is **2**, whereas the tokenized programs $\langle c, =, a, +, b \rangle$ and $\langle c, =, var, +, b \rangle$ have a edit-distance $\delta$ of **1**.

Evaluating a fixer's performance in an unsupervised setting is challenging as there are no ground truth programs to compare the output with. Ideally, the fixer should generate a program that is consistent with the user's specifications, but such specifications are not always available. Consequently, heuristics are used to evaluate the effectiveness of fixers. For instance, BIFI (Yasunaga and Liang, 2021) uses a combination of the number of bad examples that it can fix while being constrained by some edit-distance $\delta$ as the evaluation metric. In our experiments (Section 5), we demonstrate how the choice of the evaluation metric can impact the measurement of fixer's performance.

## 4 APPROACH

In this section we first briefly describe the unsupervised learning approach from BIFI (Yasunaga and Liang, 2021) to solve the syntax correction problem. We then present an overview of our new approach FILI and show how learning from the fixer's own mistakes and curriculum learning can be used to improve the *fixer*.

### 4.1 BREAK-IT-FIX-IT (BIFI)

BIFI iteratively trains two encoder-decoder models, the *breaker $B$* and the *fixer $F$*. It begins with real-world unpaired bad $X^-$ and good $X^+$ programs. In the first iteration since there is no paired data to train the *breaker* and the *fixer* models, BIFI uses synthetic data generated by heuristically perturbing the programs in $X^+$. These heuristics include randomly *a)* inserting/deleting punctuation, *b)* inserting/deleting parentheses, *c)* inserting/deleting indentations, *d)* deleting keywords (def, if, else, elif, as, return) etc.

To generate the paired synthetic data $\langle X^-_{synth}, X^+ \rangle$, BIFI selects a combination of heuristics and applies them to good programs. The resulting synthetic data is used to train the initial breaker $B_0$, which maps a good program to a bad program $B_0(X^- \mid X^+_{synth})$, and the fixer $F_0$, which maps a bad program to a good program $F_0(X^+_{synth} \mid X^-)$, in a supervised setting.

BIFI improves $B_0$ and $F_0$ through multiple rounds of the following four steps:

1. **Apply the Fixer**. $F_0$ is applied to real-world bad programs and all the programs in the predictions that parse are selected for the subsequent steps. This step generates paired real-world examples which were not available in the initial round.

2. **Fine-tune the Breaker**. $B_0$ is now fine-tuned using the paired real-world examples generated in the previous step to obtain $B_1$. This fine-tuning allows the breaker to gradually learn the real-world error distribution and generate programs that resemble real-world bad programs.

3. **Apply the Breaker**. In this step, the fine-tuned $B_1$ model is applied to real-world good programs and all the examples in the predictions that do not parse are selected for the final step. This step generates additional paired real-world-resembling examples and is used for augmenting the data generated in step 1.

4. **Fine-tune the Fixer**. Finally, $F_0$ model is fine-tuned on the paired real-world examples generated in step 1 and 3 to obtain $F_1$. This fine-tuning results in fixer trained on synthetic data to gradually learn to fix real-world bad programs. The breaker's and fixer's cyclic interaction results in both models gradually adapting to the real-world error distribution. With each iteration the performance of these models improve as they are trained on increasingly larger and more diverse datasets.

### 4.2 LEARNING FROM OWN MISTAKES (FILI)

The main distinction between BIFI and FILI lies in the breaker model and the way data is augmented in each successive round. In FILI, a single encoder-decoder model, the *fixer $F$*, is trained, and no additional breaker model training is required. In the initial step, BIFI only selects programs from the

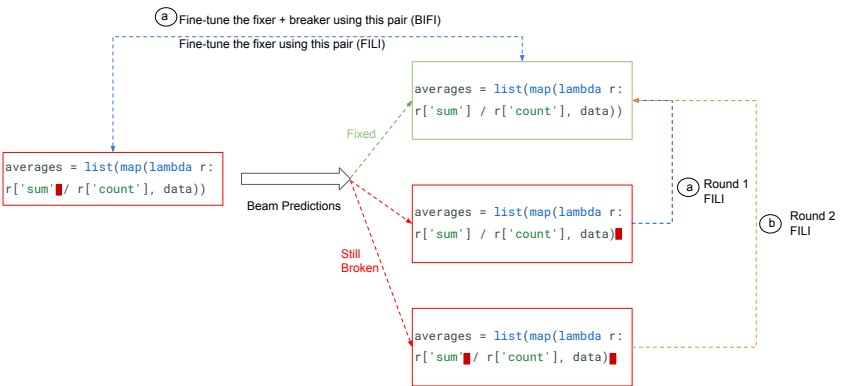

Figure 1: An illustration of how FILI's approach differs from that of BIFI. BIFI leverages only the correct programs in the beam to fine-tune the fixer (and breaker) model. In contrast, FILI in addition uses bad programs from the beam and does not require any additional breaker model for data augmentation. FILI also uses a edit-distance $\delta$ based curriculum, selecting easier pairs for training in the initial rounds and gradually introducing harder examples in the subsequent rounds.

model's predictions[1] that parse and pairs them with the incorrect source program to fine-tune the breaker and fixer models as shown in (a) in Figure 1. It completely ignores the predictions of the fixer that do not parse. In the initial rounds, these predictions account for a significant portion of the fixer's predictions as the fixer is still learning and may not have seen all the classes of errors. We observe that these incorrect programs are important, particularly those that appear higher in the beam of fixer's prediction because the fixer generates them with high confidence. As a result, there is a high likelihood that the fixer might introduce similar errors for other programs, and these programs should ideally be pushed further down in the fixer's prediction. In contrast, FILI carefully selects these programs from fixer's predictions that do not parse and pairs them with fixer's predictions that do parse to fine-tune the fixer model as shown below (a) in Figure 1.

FILI starts with the same initial fixer $F_0$ training as BIFI using heuristically generated paired synthetic data $\langle X_{synth}^-, X^+ \rangle$ to train $F_0$ in a supervised setting. FILI improves $F_0$ through multiple rounds of the following steps:

1. **Apply the Fixer**. $F_0$ is applied to real-world bad programs and all the programs in the fixer's predictions that parse are selected for the subsequent steps. This step generates paired real-world examples. Additionally, the fixer's predictions that do no parse are selected and paired with those predictions that do parse. This results in paired examples that correspond to the fixer's own mistakes.

2. **Fine-tune the Fixer**. $F_0$ in now fine-tuned on both the set of examples generated in the previous step. This fine-tuning results in a fixer that becomes more confident in fixing programs with various syntax errors over time. The negative program pairing prevents the model from generating high-scoring programs that do not parse, allowing it to effectively learn how to fix syntax errors. Essentially, the fixer is able to improve its decision boundary by decreasing the likelihood of these incorrect programs on the decision boundary, thereby improving its ability to handle various types of syntax errors with each iteration.

Our approach simplifies the data augmentation approach used in BIFI and, in essence, provides an efficient algorithm for unsupervised training of a fixer for syntax error correction. Our approach also does not require training with any additional loss functions, such as the ones used in supervised contrastive learning (Cao et al., 2021) or training a separate model to re-rank the predictions in the beam (Lee et al., 2021). As a result, FILI solves an optimization problem that is much simpler than these techniques.

---

[1]Note that when we refer to predictions, we are referring to a fixed-width beam size.

### 4.3 CURRICULUM LEARNING

Beam prediction will include many good and bad programs at varying edit-distance $\delta$ from the input incorrect source program. This provides several options to create pairs of incorrect source and good programs, as well as pairs of bad and good programs, which can be used to fine-tune the fixer. For instance, BIFI pairs good programs from the beam that are at an edit-distance of 4 from the incorrect source program. In syntax error correction, edit-distance $\delta$ can be used as a proxy to describe the *complexity* of the task. For example, fixing a program with only one syntax error is much easier than fixing a program with four syntax errors. BIFI's fine-tuning process requires the *fixer* to learn to fix all errors in a single iteration. Typically, programmers fix multiple errors iteratively by compiling, identifying, and then correcting the error. This iterative error correction resembles the curriculum learning techniques (Bengio et al., 2009), where model training is performed in a structured manner by gradually increasing the complexity of the examples, i.e., introducing easier tasks first followed by more difficult tasks.

Inspired by programmers' iterative error fixing style, FILI uses curriculum learning to improve the learning process. In Figure 1, we illustrate the process of pairing programs in each round of the FILI algorithm. In each round, we gradually increase the edit-distance of the pair of examples we use to fine-tune the fixer, where edit-distance $\delta$ denotes the number of changes the fixer makes to the incorrect program. By using edit-distance $\delta$ as a measure of complexity, we improve the fixer's ability to fix more errors in each round. The edit-distance $\delta$ criteria is used to pair both the incorrect source and good programs from the beam, as well as the bad programs and good programs from the beam (fixer's own errors). We provide details of our algorithm in Appendix A.2.

Our experiments (Section 5) indicate that curriculum learning helps improve the fixer and generates more parsable (correct) programs in the beam predictions. Our simple edit-distance $\delta$ based criteria gives insights into how neural models can be improved on programming related tasks.

## 5 EXPERIMENTS

Our approach builds over BIFI's framework (Yasunaga and Liang, 2021) and we evaluate our method on the **Github-Python** dataset collected for BIFI's evaluation.

### 5.1 MODEL AND DATASET

We use BIFI's encoder-decoder transformer architecture with 4 layers, 8 attention heads, and a hidden state size of 256 as the *fixer*. To ensure a fair comparison, we use the same initial fixer as BIFI. The initial fixer is trained on synthetic data generated by perturbing syntactically correct programs in order to introduce syntax errors (more details in Appendix A.1). We train the models on Google's TPU (v3-8). Training *fixer* for two rounds on TPU takes $\approx$ 20 hours.

We evaluate our approach on the **Github-Python** dataset[2]. The dataset consists of 38K bad programs and 3M good programs. We use the same held-out test set as BIFI i.e, from the 38k bad examples, 15k are used as the test set while the remaining 23K are available as real-world bad examples for training. Fixer's accuracy is measured by parse rate and edit-distance $\delta$. An incorrect program is considered to be fixed if the fixer's prediction parses and the edit-distance $\delta$ between the incorrect source program and the prediction is $\leq 4$ tokens. All the numbers reported are *fixer*'s top-1 accuracy. BIFI uses beam width of 10 to generate paired data for fine-tuning the fixer and the breaker, while FILI uses beam width of 30 (unless otherwise stated) to generate fine-tuning data for the fixer. For evaluating, a beam width of 10 is used for all the models.

### 5.2 RESULTS

The initial fixer (Round0) trained on synthetic data has an accuracy of **62%** on the held-out set. Since BIFI and FILI are both iterative approaches, we run two rounds for each and report the results on the held-out set in Table 1. We do not see any improvements in further rounds. All the numbers reported

---

[2]We also wanted to evaluate FILI on the DeepFix dataset, but BIFI unfortunately has not made their evaluation setup (C++ synthetic data generation, training/test splits, evaluation hyper parameters etc.) on the DeepFix dataset publicly available.

| Method | Round1 | Round2 |
|---|---|---|
| BIFI FixerOnly | 86.8% | 88.6% |
| BIFI FixerOnly⋆ | 85.1% ± 0.09% | 87.1% ± 0.45% |
| BIFI | 88.0% | 90.5% |
| **FILI Curriculum** | **89.3% ± 0.19%** | **91.2% ± 0.13%** |
| **FILI** | **89.3% ± 0.19%** | **91.6% ± 0.05%** |

| Method | Accuracy | Accuracy (edit) |
|---|---|---|
| PaLM-2 | 78.2% | 54% |
| GPT-3.5-turbo | **98.6%** | 60% |
| BIFI FixerOnly⋆ | 93.1% | 87.1% |
| BIFI | 95.5% | 90.5% |
| **FILI Curriculum** | 95.2% | **91.1%** |
| **FILI** | 96.1% | **91.6%** |

Table 1: Comparison of accuracy on the **Github-Python** dataset. Our approach FILI, utilizing *fixer*'s own mistakes for fine-tuning, outperforms BIFI FixerOnly, which relies on beam predictions but only includes programs that parse, and BIFI with the *breaker* model for data augmentation.

Table 2: Comparison of performance on the **Github-Python** dataset. Our approach FILI, outperforms over LLMs in accuracy when evaluating based on edit-distance $\delta$, and achieves comparable results when evaluating the generation of parsable programs as the evaluation metric.

in Table 1 are averaged over 5 runs. We use the following 2 configurations of BIFI and FILI for our evaluation:

1. **BIFI FixerOnly**. This configuration only fine-tunes the fixer on the bad programs it can fix in each iteration without using the the dataset generated by the breaker. It is similar to FILI as only beam predictions are used to augment data, and no breaker model is used to generate additional data.

2. **BIFI**. This configuration fine-tunes the fixer with both the bad programs it can fix and the paired examples generated by the breaker in each iteration.

3. **FILI Curriculum**. In this configuration the fixer is trained using a combination of learning from own mistakes and curriculum learning. During Round1, we use a threshold of edit-distance $\delta \leq 2$ to generate the paired data. During Round2, the paired data is generated using a threshold of edit-distance $\delta \leq 4$. Note that this is in contrast with BIFI, which uses edit-distance $\delta \leq 4$ in both rounds.

4. **FILI**. This configuration only uses learning from own mistakes to train the fixer. We use edit-distance $\delta \leq 2$ in both rounds in this configuration to generate both incorrect source and correct beam paired examples, as well as correct beam and incorrect beam paired examples.

**Note:** To eliminate the possibility that the gains observed in our results were due to changes in accelerator, we ran the BIFI FixerOnly (BIFI FixerOnly⋆ in Table 1) on TPU. The numbers reported in other two rows of BIFI correspond to those reported in the paper (Yasunaga and Liang, 2021).

We also compare BIFI and FILI against two LLMs, PaLM-2 (text-bison) (Anil et al., 2023) and GPT-3.5-turbo and report the accuracies in Table 2. We use these models in a zero-shot setting, i.e., we prompt the model with the incorrect program and ask the model to generate the corresponding correct program. The prompts used for this experiment are listed in Appendix A.6.

**Discussion.** FILI outperforms both configurations of BIFI. Compared to the FixerOnly configuration, FILI shows an improvement of **4%** in both the rounds, demonstrating that *augmenting with negative examples in addition to the positive examples from the beam helps to improve* fixer's *performance*. Moreover, FILI also outperforms the full BIFI model, which uses an additional model, breaker, to augment data in each iteration. We see an improvement of more than **1%** in both rounds. In addition to a slight improvement in performance, FILI provides a dramatically simplified training procedure as it does not require any additional model training or specialized loss functions. The breaker model in BIFI is **13 million** parameter model, and training this model requires ≈ 2days on a modest DNN accelerator. In addition to the time required for training the breaker model, the inference time for running it on the 3 million good examples also needs to be considered for each iteration. This makes BIFI's approach computationally expensive and time-consuming. In contrast, FILI only uses fixer's predictions to generate paired data leading to a much faster and more efficient training process. In relation to the overlap in problems that BIFI and FILI are capable of solving, there are certain variations. BIFI cannot solve 1263, while BIFI cannot solve 1428. Notably, there are 897 instances which both the approaches cannot solve. In terms of the unique problems that each approach can solve, FILI solves **531** unique problems, whereas BIFI solves 366 unique instances.

While our curriculum learning configuration also outperforms BIFI, we observed a slight drop in accuracy (0.5%) compared to our FILI model without the curriculum learning. Upon further analysis, we found that no curriculum learning *fixer* training aligns with the BIFI's evaluation metric (parsable and edit-distance $\delta \leq 4$) i.e. generate parsabale programs with the least number of changes to the incorrect source program. When fine-tuning the fixer with only edit-distance $\delta \leq 2$ the fixer has fewer degrees of freedom to modify the incorrect program when compared to fine-tuning with edit-distance $\delta \leq 4$. To verify this hypothesis, we relaxed the edit distance criteria for the accuracy metric while keeping the beam width fixed. Interestingly, our curriculum learning *fixer* generated **more parsable** programs than any other *fixer*. Therefore, if the objective is to train a fixer that can generate more parsable predictions in the beam, the FILI curriculum is a better configuration. We provide more details on this experiment in our ablation study (Section 5.3).

FILI (and BIFI) outperform LLMs when assessed using BIFI's evaluation metric (parsable and edit-distance $\delta \leq 4$). These models, when instructed to generate the correct program, tend to introduce modifications to other sections of the erroneous program, resulting in a higher edit-distance $\delta$ count with the incorrect program. This behavior may not be desirable for developers in real-world scenarios. When we relax the edit-distance $\delta$ criterion within the evaluation metric, we observe that GPT-3.5-turbo achieves the highest accuracy at 98.6%. However, it's worth noting that these models are trained on significantly larger datasets and possess a much higher parameter count (on the order of billions), rendering them challenging to train using commodity hardware resources. We provide detailed analysis of these experiments in Appendix A.6.

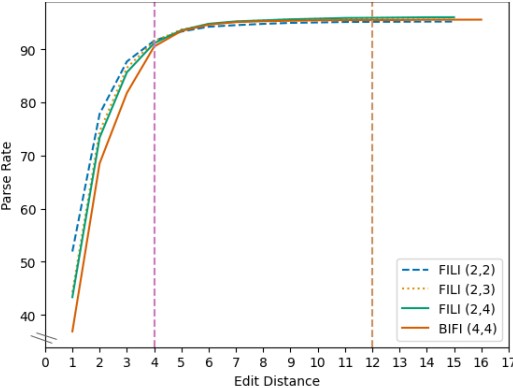

Figure 2: Comparison of parse rates for different FILI and BIFI configurations across various edit-distance $\delta$ thresholds. The line $x = 4$ represents BIFI's evaluation metric. FILI (2,2) achieves the best parse rate under this criteria as the *fixer*, indicating limited freedom for modifying the incorrect program. As the edit-distance $\delta$ threshold increase (line $x = 12$), FILI curriculum configurations ((2,3), (2,4)) show better parse rates, indicating that gradual learning improves *fixer*'s capability to generate more parsable programs.

## 5.3 ABLATION STUDY

In Section 5.2, we show that FILI outperforms all configurations of BIFI. In our ablation study, we try to answer the following questions about FILI:

1. ***Does curriculum learning generate more parsable programs?***
   In this experiment, we test our hypothesis that incorporating curriculum in fixer training can lead to higher percentage of parsable programs as the top prediction in the beam. We relax edit-distance $\delta$ criteria used for evaluation by BIFI and analyze how the accuracy of different configurations change accordingly. In Figure 2, we plot the cumulative parse rate of different configurations against the edit-distance $\delta$. The line $x = 4$ in Figure 2 represents the case where the edit-distance $\delta$ is 4, which aligns with the BIFI's evaluation metric. We observe that the FILI configurations with no curriculum (FILI (2,2)) performs the best under this criteria. However, as we move towards the right on the edit-distance $\delta$ scale (line $x = 12$), we observe that FILI configurations with curriculum generates more parsable programs compared to other configurations confirming our hypothesis. As discussed in Section 3,

| Round1 edit-distance | Round 1 | Round2 edit-distance | Round 2 |
|---|---|---|---|
| 1 | 85.5% | 4 | 91.3% |
| 2 | 89.2% | 3 | 91.5% |
| 2 | 89.2% | 4 | 91.1% |
| 2 | 89.2% | 5 | 90.8% |
| 2 | 89.2% | 6 | 90.6% |
| 4 | 89.0% | 4 | 90.0% |
| 2 | **89.3%** | 2 | **91.6%** |

Table 3: Performance comparison of FILI models with varying curricula. The *fixer* is trained with different edit-distance $\delta$ thresholds in each round.

| Method | Round1 | Round2 |
|---|---|---|
| BIFI FixerOnly⋆ | 85.1% | 87.1% |
| BIFI FixerOnly Curriculum | **86.0%** | **89%** |
| FILI Curriculum (2,4) | 89.3% | 91.2% |

Table 4: Comparison of performance between BIFI FixerOnly models trained with and without curriculum learning. Our curriculum learning approach demonstrates improved performance over the standard BIFI FixerOnly model, indicating the effectiveness of our curriculum in the training of *fixer*.

evaluating syntax fixers in unsupervised setting is challenging and the performance of the fixers can vary depending on the evaluation metric. Nonetheless, FILI models consistently outperform BIFI models across different evaluation criteria, demonstrating the effectiveness of our approach. See Appendix A.5 and Appendix A.7 for a qualitative analysis of the fixes by both the models.

2. ***How does the accuracy of* FILI *vary with different curricula?***
In this experiment, we test how the accuracy of FILI varies with different curricula. We train FILI with different edit-distance $\delta$ thresholds in each round and evaluate the model's performance using BIFI's evaluation metric (parsable and edit-distance $\delta \leq 4$). The results in Table 3 show that FILI with curriculum (2,3) performs the best amongst all the configurations. This configuration restricts the fixer the most in terms of the number of changes the fixer can make which aligns with the evaluation metric that looks for a parsable program with the minimum changes as the top prediction. As edit-distance $\delta$ threshold increases, fixer has more freedom to modify the incorrect program resulting in top programs in the beam being parsable but at a higher edit-distance $\delta$ than BIFI's evaluation metric.

3. ***Does curriculum learning help BIFI as well?***
Next, we test whether curriculum learning can also benefit BIFI's FixerOnly configuration. We train the FixerOnly model with our curriculum (2,4) and evaluate it's performance using BIFI's evaluation metric. As shown in Table 4, we observe a 1% improvement in round1 and a 2% improvement in round2 compared to the original FixerOnly model. These reuslts indicate that incorporating curriculum learning can indeed improve the learning process for BIFI as well. This experiment demonstrates the effectiveness of our curriculum learning approach in improving the performance of not only the FILI model but also the BIFI model. Furthermore, since the FixerOnly configuration is similar to FILI without the negative programs pairing, these results indicate the importance of incorporating negative programs from the beam in the training process. The inclusion of these programs allows the model to improve it's ability to fix different classes of errors.

## 6 CONCLUSION

In this paper, we presented FILI, a new approach for automatically learning a syntax-error fixer in an unsupervised setting. In contrast to prior approaches (Yasunaga and Liang, 2021) that rely on separate models to augment data in each iteration, our method simplifies this step significantly. We leverage the model's own mistakes in its predictions to augment data and we do this in a curriculum style approach by gradually increasing the complexity of these examples in each iteration. Our evaluation demonstrates that our approach results in a fixer that outperforms the prior approaches while significantly reducing the training time. Our approach opens up new research directions in unsupervised training of models for programming-related tasks.

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
