# A    APPENDIX

## A.1    DATASET

In order to perform a fair comparison against BIFI, we adopt the exact same setting in terms of synthetic data generation and the model architecture as employed by BIFI. We use BIFI's code for generating the training data and establishing the data splits to ensure consistency throughout the comparison. BIFI uses two approaches for generating the synthetic data:

1. **Random noising**: in this tokens in syntactically correct programs are randomly dropped, inserted or replaced to generate a syntactically incorrect program.

2. **Heuristic nosing**: this involves careful selection of tokens to drop/insert/replace to insert syntax errors. Few of these include randomly dropping/inserting/replacing:

   (a) Operators (Boolean, arithmetic, relational) and punctuation's (:, ,(),[])
   (b) Identifiers
   (c) Keywords
   (d) Identifier Types

The test dataset used for evaluation constain 15055 incorrect examples. Within this dataset, 3999 examples have unbalanced parentheses errors, 6307 have indentation errors and 4749 have other syntax errors including missing colon, missing comma, missing newline, redundant comma, invalid use of common, among others.

## A.2    FILI ALGORITHM

---
**Algorithm 1** Curriculum Learning with FILI

---
**Require:** Real-world Incorrect program: a set of examples $x_1, x_2, \ldots, x_n$ where each $x_i$ is an incorrect program; number of rounds $num\_rounds$; maximum edit distance $d_{\max}$
1: $finetune \leftarrow \emptyset$
2: $d \leftarrow 1$
3: **for** round $r = 1$ to $num\_rounds$ **do**
4:     **for** each $x_b \in x_1, x_2, \ldots, x_n$ **do**
5:         $Pred \leftarrow$ Apply *fixer* to $x_b$ and get a set of candidate corrections $Pred$
6:         **for** each $p \in Pred$ **do**
7:             **if** $\mathcal{C}\{p\}$ and $\delta(p, x_b) \leq d$ **then**
8:                 $finetune \leftarrow finetune \cup \langle x_b, p \rangle$
9:                 Add $p$ to the set of correct programs $correct$
10:             **else**
11:                 Add $p$ to the set of incorrect programs $incorrect$
12:             **end if**
13:         **end for**
14:     **end for**
15:     **for** each $i \in incorrect$ **do**
16:         **for** each $c \in correct$ **do**
17:             $finetune \leftarrow finetune \cup \langle i, c \rangle$
18:         **end for**
19:     **end for**
20:     $d \leftarrow d + 1$
21:     Fine-tune *fixer* on the pairs in $finetune$
22: **end for**
23: **return** *fixer*

---

Algorithm 1 outlines the FILI framework. The algorithm is used to fine-tune the fixer that was initially trained on synthetic data. The fine-tuning process is iterative and involve multiple rounds. In each round, we apply the fixer to the set of real-world incorrect programs to generate a set of candidate predictions (line 5). From this set, we select the examples that parse and are at a current maximum edit-distance $\delta$ from the incorrect source program (lines 7-9). We add these examples to

the fine-tuning set and to the set of correct programs. If the prediction does not parse, we add it to the set of incorrect programs (line 11). Next, we extend our fine-tuning set by pairing incorrect examples from the beam with the correct predictions (lines 15-19). Finally, we increment our maximum edit-distance $\delta$ (line 20) and fine-tune the fixer (line 21) on the set of examples collected in this round.

## A.3 BEAM SIZE ABLATION

| Method | Round1 | Round2 |
|--------|--------|--------|
| FILI 10 | 88.4% | 91.3% |
| FILI 25 | 89.4% | 91.5% |
| FILI 30 | 89.2% | 91.6% |
| FILI 50 | 89.3% | 91.5% |

Table 5: Beam Size Ablation

FILI uses fixer's predictions in the beam to generate data for fine-tuning. In this experiment we test the performance of FILI with varying beam width sizes as the size of the beam can influence the number of examples which are generated for fine-tuning. For this experiment, we use our best FILI configuration (2,2) which uses edit-distance $\delta \leq 2$ for both the rounds. From Table 6, we observe that FILI models are fairly robust to the beam-width size. As we keep increasing the beam size (from 10 to 30) we observe accuracy gains in both the rounds. However, at beam width 50, we observe that the performance saturates. We hypothesize that the predictions that lower in the beam do not add any useful learning signal

## A.4 ADDITIONAL TRAINING DATA

In Table 6, we provide the number of examples from the beam which are used as additional data for fine-tuning. These numbers include both incorrect source programs along with the good program from the beam pair, as well as the bad program and the good programs from the beam pair. In contrast, BIFI uses as additional breaker model to augment the training data for the fixer.

| Method | #Examples Round1 | #Examples Round2 |
|--------|------------------|------------------|
| FILI (2,2) | 16,97,175 | 9,07,991 |
| FILI (2,4) | 16,97,175 | 17,52,568 |

Table 6: The number of additional training examples different configurations of FILI uses in each round. Note that this is significantly less than BIFI which applies the trained breaker model on the 3 million correct examples which are then subsequently used to train the fixer model.

## A.5 QUALITATIVE EXAMPLES

In Figure 3, we show instances from the test set which our model, FILI, is able to fix while BIFI's top prediction is incorrect. In the first example, even though the BIFI fixed the error by placing the ',' at the error location but the model made changes to the next line of code introducing other errors in the program. In the second example, BIFI completely misses out on fixing the indentation errors while our model is able to fix generate the correct indentations for both the lines of code.

In Figure 4, we show instances from the test set which our, FILI, is not able to fix while BIFI's top prediction is correct. In the first example, FILI fixes the error in the first line of code by removing the extra parenthesis but introduces another error in the last line of the code by removing a closing parenthesis. In the second example, our model adds the 'pass' keyword at the error location but completely misses out on the indentation errors on the next few lines.

It should be noted that these examples are top predictions from both the FILI and BIFI models. One thing we observe is that they consistently identify the location of the error correctly. While FILI performs better in certain instances, BIFI performs better on others. One interesting possibility for future exploration would involve utilizing FILI for the initial training rounds and then incorporating

| Incorrect Source | BIFI Prediction (Incorrect) | FILI Prediction (Correct) |
|---|---|---|
| ```python
"str"
import numpy
from pyscf import gto
from pyscf . pbc import gto as
pgto
cell = pgto . M (
    atom = "str" ,
    basis = { "str" : gto . parse
( "str" ) , } ,
    pseudo = "str" ,
    h = numpy . eye ( 3 ) * 3.5668
,
    gs = [ 10 ] * 3 )
``` | ```python
import numpy
from pyscf import gto
from pyscf . gto import pgto as
pgto
cell = gto . M (
    atom = "str" ,
    basis = { "str" : pseudo .
parse ( "str" ) , } ,
    3.5668 = "str" ,
    h = numpy . eye ( 3 ) * 3.5668
,
    gs = [ 10 ] * 3 )
``` | ```python
"str"
import numpy
from pyscf import gto
from pyscf . REPO_ROOT import gto
as pgto
cell = pgto . M (
    atom = "str" ,
    basis = { "str" : gto . parse
( "str" ) } ,
    pseudo = "str" ,
    h = numpy . eye ( 3 ) * 3.5668
,
    gs = [ 10 ] * 3 )
``` |
| ```python
def listTickets ( self , type ) :
    "str"
    get_params = self .
generateParams ( )
    ret = self . getDataAPI (
type . strip ( ) , get_params )
    print ( self .
printJson ( ret ) )
``` | ```python
def listTickets ( self , type ) :
    "str"
    get_params = self .
generateParams (
    ret = self . getDataAPI (
type . strip ( ) , get_params ) ,
    print ( self . printJson (
ret ) ) )
``` | ```python
def listTickets ( self , type ) :
    "str"
    get_params = self .
generateParams ( )
    ret = self . getDataAPI ( type
. strip ( ) , get_params )
    print ( self . printJson ( ret
) )
``` |

Figure 3: Instances from the held-out test set on which BIFI's top prediction is incorrect while FILI's top prediction is correct.

| Incorrect Source | FILI Prediction (Incorrect) | BIFI Prediction (Correct) |
|---|---|---|
| ```python
def syntax_error ( self ( ( token ,
text ) ) :
    if token == - 1 :
        self . errors . append (
"str" % ( text , ) )
    else :
        self . errors . append (
"str" % (
        text , "str" . join ( [
str ( t ) for ( _t , t ) in self .
context [ : - 1 ] ] ) ) )
``` | ```python
def syntax_error ( self , token ,
text ) :
    if token == - 1 :
        self . errors . append (
"str" % ( text , ) )
    else :
        self . errors . append (
"str" % (
        text , "str" . join ( [
str ( t ) for ( _t , t ) in self .
context [ : - 1 ] ] ) )
``` | ```python
def syntax_error ( self , token ,
text ) :
    if token == - 1 :
        self . errors . append (
"str" % ( text , ) )
    else :
        self . errors . append (
"str" % (
        text , "str" . join ( [
str ( t ) for ( _t , t ) in self .
context [ : - 1 ] ] ) ) )
``` |
| ```python
def p_error ( p ) :
    if p :
    print ( p )
    exit ( )
    else :
        print ( "str" )
        exit ( )
``` | ```python
def p_error ( p ) :
    if p : pass
    print ( p )
    exit ( )
    else :
        print ( "str" )
        exit ( )
``` | ```python
def p_error ( p ) :
    if p :
        print ( p )
        exit ( )
    else :
        print ( "str" )
        exit ( )
``` |

Figure 4: Instances from the held-out test set on which FILI's top prediction is incorrect while BIFI's top prediction is correct.

BIFI's breaker model approach in later rounds. This combined approach could potentially enhance the fixer's predictions even further.

```
def Check ( serial ) :          def main ( ) :               def backtracking ( c ) :
    "str"                           x = None                     if reject ( P , c ) then return
    must serial                     a = 2                        if accept ( P , c ) then output ( P , c )
    site = serial . split ( "str" ) [ 0    switch a :            s < - first ( P , c )
]                                       case 1 :                 while s != ^ :
    d2 = . d . get ( site )                x = "str"              backtracking ( s )
    if d2 :                             case 2 :                  s < - next ( P , s )
        if serial in d2 :                  x = "str"
            return False                default :
        if serial <= d2 [ "str" ] :        break
            return False            TestError ( x == "str" )
    . _Add ( serial )
    return True
```

Figure 5: Examples showing programs from the test-set which are not fixed by either BIFI or FILI. These programs require inserting program statements or deleting some of the statements to fix the syntax errors. BIFI and FILI are not trained to perform such operations.

| Model | $\delta = 4$ | $\delta = 6$ | $\delta = 8$ | $\delta = 10$ | $\delta = \inf$ |
|---|---|---|---|---|---|
| PaLM-2 | 54% | 67.4% | 73% | 75.5% | 78.2% |
| GPT-3.5-turbo | 60% | 70.8% | 78% | 83.3% | 98.6% |

Table 7: Accuracy of the LLMs on the held-out test set with varying edit-distance $\delta$ criterion.

### A.5.1 PROGRAMS NOT FIXED

We performed a manual inspection of the examples that both BIFI and FILI cannot fix. We observed that some of the problems are not purely syntax repair problems. We provide examples of some of these programs from the test-set in Figure 5. For instance, some of these programs are incomplete and require generating additional program statements to fix them. Similarly, some of these programs require deleting certain program statements which neither FILI or BIFI is trained to perform. Finally, we observed some of the programs were written in other languages such as C++ and Java. This observation indicates the fact that the upper limit for success on this test set is not 100%, and the test set would require a careful analysis to determine how many programs are actually fixable. Consequently, 1% improvement over BIFI might indeed be reasonable within this context.

### A.6 LLM EXPERIMENTS

**Experiment Details.** We use PaLM-2 (text-bison) and GPT-3.5-turbo for our experiments. We evaluate these models in the zero-shot setting. In Figure 6, we show the prompt we use to evaluate these models. We set the temperature $= 0$ to avoid any randomness from the models.

```
The following python program has syntax
errors. Fix all the syntax errors to make the
program parsable.
<Incorrect Program>:
```
{{incorrect}}
```
<Correct Program>:
```

Figure 6: Prompt used for evaluating the LLMs.

**Evaluation.** In our evaluation, we use the same evaluation metric as (Yasunaga and Liang, 2021), where the predicted program should be parsable, and the edit-distance $\delta$ distance between the incorrect source program and the prediction should be $\leq 4$ tokens. As discussed in Section 3, particularly in an unsupervised context where we lack access to ground-truth correct programs, this edit-distance $\delta$

threshold plays a crucial role in ensuring that the models minimize changes to parts of the incorrect program beyond the actual syntax errors. In Table 7, we present the accuracy results for various edit-distance $\delta$ thresholds, with a more flexible edit-distance $\delta$ constraint applied to LLMs predictions, as these models are not specifically trained for the syntax fixing task. Even when allowing an edit-distance $\delta \leq 10$, our model FILI consistently outperforms GPT-3.5 and PaLM-2 by $\approx$**8%** and $\approx$**16%** respectively. If we further relax the edit-distance $\delta$ criteria to focus solely on generating parsable programs, we observe that GPT-3.5 achieves an accuracy of 98.6%, while FILI achieves an accuracy of 96.1%. It is important to note that these numerical comparisons are provided for reference, and in practical, real-world settings, users may not favor a model that introduces significant modifications to their code. Nonetheless, it's worth noting that FILI outperforms PaLM-2, despite the latter having at least 100 times more parameters and being trained on significantly larger datasets compared to our model.

**Qualitative Analysis**. In Figure 8 and Figure 9, we provide illustrations of incorrect programs and the corresponding predictions generated by GPT-3.5 and PaLM-2, respectively. These figures show instances where both models introduce changes to other parts of the incorrect program, resulting in a program at greater edit-distance $\delta$ from the original incorrect source program. It is important to note that in all these instances, both models effectively fix the syntax error. For instance, in the initial example depicted in Figure 8, the simplest correction would involve removing the parentheses around "r, g, b," however the model suggests a solution of combining them into a single variable and subsequently unpacking it. Such recommendations may not align with user preferences, as modifying the function's definition may necessitate changes to other portions of the program that invoke the function. Additionally, these models occasionally introduce stylistic changes to the code, such as eliminating inlining within an if-else block. These changes may not be related to program errors and may not align with user preferences, especially when dealing with small code snippets that are part of a larger codebase adhering to specific formatting standards.

LLMs are general-purpose models trained on extensive code datasets, enabling them to handle a wide array of programming-related tasks, including code generation, summarization, and translation, among others. However, they still make mistakes when applied to specific tasks such as syntax error correction, primarily due to not being specialized for these particular tasks. This limitation becomes evident from the accuracy results for the PaLM-2 model in Table 2. We illustrate examples in Figure 10 and Figure 11, where both models fail to fix syntax errors in certain test-set examples. In other words, the predictions generated by the models still contain syntax errors. For instance, in the second example depicted in Figure 10, while the model correctly fixes the parentheses errors in the program, it misses the colon on line 3.

**Comparison with FILI and BIFI**. Given GPT-3.5-turbo's notably high parse rate among all the models, we compare its predictions with those generated by FILI and BIFI. Specifically, we focus on GPT-3.5's predictions that align with the correctness criteria outlined in BIFI's evaluation metric. From Table 7, GPT-3.5 has an accuracy rate of 60% (9033 programs). Our analysis involves identifying the intersection of problems solvable by all three models: GPT, BIFI, and FILI (682 programs). Interestingly, we observe that FILI's predictions align precisely with GPT's predictions in **44%** of these cases, while BIFI achieves a **41%** match rate. This observation further highlights the quality and effectiveness of FILI's predictions.

## A.7 DISCUSSION ON EVALUATION METRIC

An ideal evaluation criteria would be whether after performing the fix to the incorrect program, does the program satisfies the user specification. There are several challenges to using this criteria but the two major problems are:

1. In this work, we are just fixing the syntax errors in the program. So, even if the specification is given and we fix the syntax errors, the program may still not satisfy the specification as it may have other semantic errors which are beyond the scope of this work

2. The dataset used for evaluation does not have any form of user specification or assertions that the program should satisfy, and it is also not always possible to run these programs as in most of the cases these are small snippets of code extracted from a large codebase and cannot be run in isolation.

Figure 7: Examples showing incorrect programs from the test-set that require more than 4 edits to fix the syntax errors. The left-hand-side shows the incorrect program and the right-hand-side shows FILI's predictions.

Therefore, designing an evaluation metric for syntax repair is a challenging research problem and we provide a discussion of this in Section 3 of the paper. Edit distance is commonly used in the program repair evaluation to ensure that the repair systems do not make arbitrary changes to the program. To make a fair comparison against BIFI, the previous state-of-the-art model, we used the exact same evaluation metric as BIFI based on the edit-distance. We also show in our experiments how changing this metric can lead to changes in the fixer performance. In Figure 7, we show examples from the test-set which require more than 4 (BIFI's edit-distance $\delta$ threshold for correctness) and the corresponding fix generated by FILI. Even though FILI generates the correct solution, these programs will be counted as not correct under BIFI's evaluation metric as these violate the edit-distance $\delta$ threshold. Nonetheless, for all the metrics used in the paper for evaluation, we observe that FILI performs better than BIFI.

```
1 def darkenRGB((r, g, b), factor):
2     '''Darkens a given RGB color value by a factor between 0 and 1'''
3     maxColor = 255
4     r *= 1 - factor
5     g *= 1 - factor
6     b *= 1 - factor
7     r, g, b = int(round(r)), int(round(g)), int(round(b))
8     return(r, g, b)
```

```
1 def darkenRGB(rgb, factor):
2     '''Darkens a given RGB color value by a factor between 0 and 1'''
3     maxColor = 255
4     r, g, b = rgb
5     r *= 1 - factor
6     g *= 1 - factor
7     b *= 1 - factor
8     r, g, b = int(round(r)), int(round(g)), int(round(b))
9     return (r, g, b)
```

```
1 def eTargetMissile(self, sender, (x, y)):
2
3     if self.eOutLaunchMissile:
4         self.eOutLaunchMissile(self.targetting, (x, y))
5     self.targetting = None
```

```
1 def eTargetMissile(self, sender, coordinates):
2     x, y = coordinates
3     if self.eOutLaunchMissile:
4         self.eOutLaunchMissile(self.targetting, coordinates)
5     self.targetting = None
```

```
1 def create_word_sentiments(pos_tweets, neg_tweets):
2     for(words, sentiment) in pos_tweets + neg_tweets:
3         words_filtered = [e.lower() for e in words.split() if len(e) >= 3]
4         tweets.append((words_filtered, sentiment)
5
```

```
1 def create_word_sentiments(pos_tweets, neg_tweets):
2     tweets = []
3     for (words, sentiment) in pos_tweets + neg_tweets:
4         words_filtered = [e.lower() for e in words.split() if len(e) >= 3]
5         tweets.append((words_filtered, sentiment))
6     return tweets
```

```
1 def getQueueSystem(self, test):
2     queueModuleText = queueSystemName(test)
3     if queueModuleText is None:
4         return None
5     queueModule = queueModuleText.lower()
6     if self.queueSystems.has_key(queueModule):
7         return self.queueSystems[queueModule]
8     command = "from " + queueModule + " import QueueSystem as _QueueSystem"
9     exec command
10    system = _QueueSystem(test)
11    self.queueSystems[queueModule] = system
12    return system
13
```

```
1 def getQueueSystem(self, test):
2     queueModuleText = queueSystemName(test)
3     if queueModuleText is None:
4         return None
5     queueModule = queueModuleText.lower()
6     if queueModule in self.queueSystems:
7         return self.queueSystems[queueModule]
8     command = "from " + queueModule + " import QueueSystem as _QueueSystem"
9     exec(command)
10    system = _QueueSystem(test)
11    self.queueSystems[queueModule] = system
12    return system
```

```
1 def spiral(n, a):
2     thetas = map(
3         lambda n: pi / 5 + pi / 20 * n,
4         xrange(n))
5     rs = thetas
6     return map(
7         lambda(r, theta): (r * cos(theta + a), r * sin(theta + a)),
8         zip(rs, thetas))
9
```

```
1 def spiral(n, a):
2     from math import pi, cos, sin
3     thetas = list(map(
4         lambda n: pi / 5 + pi / 20 * n,
5         range(n)))
6     rs = thetas
7     return list(map(
8         lambda r_theta: (r_theta[0] * cos(r_theta[1] + a), r_theta[0] * sin(r_theta[1] + a)),
9         zip(rs, thetas)))
```

Figure 8: Examples showing GPT-3.5-turbo model making changes to other parts of the incorrect program. The left-hand-side of the figure shows the incorrect programs and the right-hand-side shows the predictions from the model.

```
1 def LCS(A, B):                          1
2     m = len(A)                          2 def LCS(A, B):
3     n = len(B)                          3     m = len(A)
4     c = [, ]                            4     n = len(B)
5     d = [, ]                            5     c = [[0 for i in range(n + 1)] for j in range(m + 1)]
6                                         6     d = [[0 for i in range(n + 1)] for j in range(m + 1)]
                                          7
                                          8
```

```
1 class MagicError(Exception):                                          1
2     def __init__(self, error, description):                           2 class MagicError(Exception):
3         self.response, self.content = response, content              3     def __init__(self, response, content):
4     def __str__(self):                                                4         self.response, self.content = response, content
5         return "MudpuppyError: %s: %s" %(str(self.response), str(self.content))   5     def __str__(self):
6                                                                       6         return "MagicError: %s: %s" %(str(self.response), str(self.content))
                                                                        7
                                                                        8
```

```
1 def plt_signal(x, y, labelx, labely, direccion, mostrar, guardar)     1
2     '''Plots and saves a signal.'''                                  2 def plt_signal(x, y, labelx, labely, direccion, mostrar, guardar):
3     plt.plot(x, y)                                                   3     '''Plots and saves a signal.'''
4     plt.plot(times, envolv, linewidth = 3)                           4     plt.plot(x, y)
5     plt.xlabel('tiempo (s)')                                         5     plt.plot(times, envolv, linewidth=3)
6     plt.ylabel('amplitud')                                           6     plt.xlabel('tiempo (s)')
                                                                        7     plt.ylabel('amplitud')
7     plt.show()                                                       8     if mostrar:
                                                                        9         plt.show()
8     plt.savefig(direccion + '.png')                                 10     if guardar:
9     plt.close()                                                     11         plt.savefig(direccion + '.png')
10    pass                                                            12     plt.close()
11                                                                    13     pass
                                                                       14
                                                                       15
```

```
1 def int2b128(integer, stream):                                       1
2     if integer == 0:                                                 2 def int2b128(integer, stream):
3         stream(chr(0)                                                3     if integer == 0:
4         return                                                       4         stream.write(chr(0))
5     assert integer > 0, "can only encode positive integers"          5         return
6     while integer:                                                   6     assert integer > 0, "can only encode positive integers"
7         stream(chr(integer & 0x7f))                                  7     while integer:
8         integer = integer >> 7                                       8         stream.write(chr(integer & 0x7f))
9                                                                      9         integer = integer >> 7
                                                                       10
                                                                       11
```

```
1 def do_search():                                                     1
2     key = request.GET.get('keyword')                                 2 def do_search():
3     key = key.strip()                                                3     key = request.GET.get('keyword')
4     form = SearchForm()                                              4     key = key.strip()
5     if key != None or len(key) < 3:                                  5     form = SearchForm()
6     posts = Post.query(Post.body.search(key))                        6     if key != None and len(key) < 3:
7     return{"posts": posts, "form": form, "key": key}                 7         posts = Post.query(Post.body.search(key))
8     return "Hello"                                                   8         return {"posts": posts, "form": form, "key": key}
9                                                                      9     return "Hello"
                                                                       10
                                                                       11
```

Figure 9: Examples showing PaLM-2 model making changes to other parts of the incorrect program. The left-hand-side of the figure shows the incorrect programs and the right-hand-side shows the predictions from the model.

```
n
                                                            n  1 import random
                                                               2 import math
                                                               3
1 def randomize((x, y), rx, ry, norm):                      4 def randomize((x, y), rx, ry, norm):
2     if norm:                                              5     if norm:
3         r = abs(random.normalvariate(0.0, 0.5 * max(rx, ry)))  6         r = abs(random.normalvariate(0.0, 0.5 * max(rx, ry)))
4     else:                                                 7     else:
5         r = random.uniform(0.0, max(rx, ry))              8         r = random.uniform(0.0, max(rx, ry))
6     a = random.uniform(0.0, 2 * math.pi)                  9     a = random.uniform(0.0, 2 * math.pi)
7     x += math.cos(a) * rx                                10     x += math.cos(a) * rx
8     y += math.sin(a) * ry                                11     y += math.sin(a) * ry
t 9   return[x, y]                                         t12     return [x, y]
10
```

```
f 1 def set_default(ini):                    f 1 def set_default(ini):
t 2     info = init.Info('{0}.ini'.format(ini) t 2     info = init.Info('{0}.ini'.format(ini))
  3     with open(info.init_file, 'w'(          3     with open(info.init_file, 'w')
  4
```

```
n 1 '''find_unused_options.py'''           n 1 find_unused_options.py'''
  2 import argparse                           2 import argparse
  3 import os                                 3 import os
  4 import sys                                4 import sys
n 5 sys.path.append(os.getcwd()             n 5 sys.path.append(os.getcwd())
  6 from oslo.config import iniparser         6 from oslo.config import iniparser
t 7                                         t
```

```
f 1 def getHorizDir((x1, y1, z1), (x2, y2, z2)): f 1 def getHorizDir((x1, y1, z1), (x2, y2, z2)):
  2     if abs(x2 - x1) > abs(z2 - z1):          2     if abs(x2 - x1) > abs(z2 - z1):
n 3         return(sign(x2 - x1), 0, 0)        n 3         return (sign(x2 - x1), 0, 0)
  4     else:                                    4     else:
n 5         if(z2 == z1):                      n 5         if z2 == z1:
  6             return(1, 0, 0)                  6             return (1, 0, 0)
  7         else:                                7         else:
t 8             return(0, 0, sign(z2 - z1))   t 8             return (0, 0, sign(z2 - z1))
  9
```

Figure 10: Examples showing incorrect programs that GPT-3.5-turbo model was not able to fix. The left-hand-side of the figure shows the incorrect programs with the syntax error marked in red circle and the right-hand-side shows the incorrect predictions from the model.

Figure 11: Examples showing incorrect programs that PaLM-2 model was not able to fix. The left-hand-side of the figure shows the incorrect programs with the syntax error marked in red circle and the right-hand-side shows the incorrect predictions from the model.