# OpenReview forum: "FILI: Syntax Repair By Learning From Own Mistakes"
_ICLR.cc/2024/Conference — Submitted to ICLR 2024_

### Official Review · Reviewer_ydVV · 2023-10-26

**Soundness:** 3 good
**Presentation:** 4 excellent
**Contribution:** 2 fair
**Rating:** 6
**Confidence:** 4

**Summary:**

The paper proposes Fili, an to training syntax repair neural networks. Fili works by iteratively attempting to repair a set of broken programs, then adding all pairs of (original broken program, fixed proposed program) and (still broken proposed program, fixed proposed program) to its dataset, re-training on the new dataset, and repeating. The paper augments this by proposing a curriculum learning approach, in which the neural network is presented with (broken, fixed) program pairs with larger and larger edit distances over the course of training. The paper evaluates Fili against the prior state-of-the-art baseline, Bifi, and finds that Fili improved on Bifi by about 1%. The paper also evaluates Fili against LLMs prompted to perform syntax repair, finding that GPT-3.5 generates parseable programs more often than Bifi, but with a significantly higher edit distance (making other program changes).

**Strengths:**

* The problem domain is interesting and well motivated
* The solution itself (Fili) is clever, and leads to a simpler training approach than prior work
* In addition to being simpler, the proposed approach also performs somewhat better than prior work (BiFi).
* The paper is quite well written: I had no issues understanding any content or concepts
* The evaluation is fairly extensive, comparing a range of baselines, ablations, and other related research questions

**Weaknesses:**

* The intuition of the connection between iterative error fixing and curriculum learning (Section 4.3) is tenuous at best
* The evaluation shows only modest improvements compared to prior work, and is potentially outperformed by LLMs:
  * Fili uses a beam width of 30, while Bifi uses a beam width of 10. Bifi is a somewhat more involved model though. Are the FLOPs used to train equivalent between these models? I do see that Appendix A.3 has Fili with a beam size of 10: why was this not chosen as the model evaluated in the paper (for fairness with Bifi)?
  * The LLM experiments are zero-shot and do not include GPT-4, but still surpass the proposed approach in the accuracy (without edit distance) metric. As for accuracy with edit distance, the LLM's prompt ("Fix all the syntax errors to make the program parsable") does not include the statement that the program should remain otherwise unchanged or that the edit distance should be minimized.

**Questions:**

* What is the comparison in #parameters and #FLOPs of the BiFi and Fili models in the evaluation?
* Do LLMs still result in a large edit distance when examples are provided in the prompt, or when the prompt is modified to mention that the program should remain otherwise identical?

---

> ### Author Response · Authors · 2023-11-21
>
> **comparison in #parameters and #FLOPs of the BiFi and Fili models and beam widths**
>
> To perform a fair comparison with BIFI, we adopt the exact same settings in terms of synthetic data generation and the model architecture, as employed by BIFI. Consequently, the number of parameters for the fixer (encoder-decoder) is identical for both BIFI and FILI. However, as we skip training the breaker model, we reduce the total number of parameters by half compared to BIFI.
>
> Since FILI relies on the beam for generating additional data, we use a slightly higher beam-width than BIFI during inference. This adjustment is specific to the inference process, and the impact of beam-width on FLOPs is relatively small compared to the full breaker model used by BIFI for generating additional data. Overall, FILI exhibits significantly lower FLOPs than BIFI, as it avoids training a complete 13 million parameter model.
>
> **GPT-4 Experiments:**
>
> Due to budget constraints, we could not run inference on the full test-set (15055 examples) using GPT-4. For comparison, we run a small scale experiment by randomly sampling 2000 programs from the test-set and prompting the model in zero-shot setting using the following prompt:
> The following python program has syntax errors. Fix all the syntax errors to make the program parsable. Do not make any other changes to the program. Just fix the syntax errors.
>
> '''
>
> <Incorrect Program>:
> ```
> {{incorrect}}
> ```
> <Correct Program>:
>
> '''
>
> Results:
> | Model | Edit = 4 | Edit = 6 | Edit = 8 | Edit = 10 | Edit = inf |
> |-------|----------|----------|----------|-----------|------------|
> | GPT-4 | 57.3%    | 73.25%   | 80.95%   | 86.95%    | 99.75%     |
>
> The observations from this experiment align with those using other LLMs. The model shows good performance in fixing syntax errors but introduces several changes to other parts of the program, which might be undesirable in real-world scenarios. Furthermore, both the training and inference processes of these models are orders of magnitude more computationally expensive (and even costly) compared to our model.

---

### Official Review · Reviewer_yQj5 · 2023-10-28

**Soundness:** 3 good
**Presentation:** 3 good
**Contribution:** 2 fair
**Rating:** 5
**Confidence:** 3

**Summary:**

This work proposes a FILI (fix-it-learn-it) method to train the model for fixing syntax errors in programs. It improves over the existing BIFI approach, without having to train any additional models for data augmentation. As a result, in each iteration, FILI finetunes the fixer model by its own prediction.

**Strengths:**

The proposed FILI method appears to be reasonable, and it simplifies the data augmentation approach used in BIFI. Thereby, the new method is much more efficient and easier in the training, and it achieves (slightly) better results than BIFI.

**Weaknesses:**

The delta-distance based metric adopted in the evaluation cannot fully reflect the repair performance when comparing with repair baselines. To justify that FILI outperforms large language models (LLMs), the edit accuracy subject to some delta-distance is used, with δ denoting the number of changes the fixer makes to the incorrect program.  It turns out this edit accuracy is inadequate and potentially biased, as it overlooks the semantic correctness of the program and it also ignores the possible semantic change after the repair.


Syntax errors are a class of relatively easier software problems to repair, and it seems that LLMs handle program syntax repair even better than FILI regarding accuracy.  It was mentioned that LLMs tend to make more changes in the program repair. However, the changes made by LLMs may depend on how the LLMs were prompted.

**Questions:**

Is there a way to more comprehensively compare with FILI and LLMs for repairing program syntax?

---

> ### Author Response · Authors · 2023-11-21
>
> **Is there a way to more comprehensively compare with FILI and LLMs for repairing program syntax?**
> We agree with your comment that this is not the ideal evaluation criteria. An ideal evaluation criteria would be whether after performing the fix to the incorrect program, does the program satisfies the user specification. There are several challenges to using this criteria but the two major problems are:
> 1. In this work, we are just fixing the syntax errors in the program. So, even if the specification is given and we fix the syntax errors, the program may still not satisfy the specification as it may have other semantic errors which are beyond the scope of this work.
>
> 2. The dataset used for evaluation does not have any form of user specification or assertions that the program should satisfy, and it is also not always possible to run these programs as in most of the cases these are small snippets of code extracted from a large codebase and cannot be run in isolation.
>
> Therefore, designing an evaluation metric for syntax repair is a challenging research problem and we provide a discussion of this in Section 3 of the paper. Edit distance is commonly used in the program repair evaluation to ensure that the repair systems do not make arbitrary changes to the program. To make a fair comparison against BIFI, the previous state-of-the-art model, we used the exact same evaluation metric as BIFI based on the edit-distance. We also show in our experiments how changing this metric can lead to changes in the fixer performance. Nonetheless, for all the metrics used in the paper for evaluation, we observe that FILI performs better than BIFI.

---

> > ### Comment · Reviewer_yQj5 · 2023-11-23
> >
> > Thank you for the reply. I am not changing my score. But I tend to think that FILI has advantages vs LLMs, which is subject to better justification.

---

### Official Review · Reviewer_5pCb · 2023-10-30

**Soundness:** 3 good
**Presentation:** 3 good
**Contribution:** 3 good
**Rating:** 5
**Confidence:** 4

**Summary:**

The paper presents a new technique for repairing syntax errors. The goal is to train a fixer model, one that takes a "bad program" that does not parse, and outputs a "good program" that parses correctly.

To control for arbitrary modifications, evaluation only considers edits that are up to some fixed edit distance from the original "bad" program.

Previous work (BIFI) used a combination of a Fixer/Breaker models to  train the Fixer model (and simultaneously, a Breaker model used to generate incorrect examples that are similar to real-world bad programs).

In contrast, this work (FILI) only uses a single Fixer model, and uses negative samples from the Fixer's beams to augment the data used to fine-tune the fixer. FILI uses high-confidence incorrect predictions from the highly-ranked beams as negative examples to be paired with the correct program (one that parses). This is similar to the approach taken by (Cao et al. 2021).

The new approach shows a modest improvement over BIFI, but does that when only using a single Fixer model.

**Strengths:**

- Thorough evaluation. Appreciated the supplemental materials and the qualitative examples. These were very helpful, especially with respect to the evaluation metric.

- It is surprising and valuable to note that a FILI outperforms BIFI while only using a single Fixer model, leveraging negative samples from the Fixer's beams. Maybe this says something about the nature of the errors being fixed and how close they are to the correct program?

**Weaknesses:**

- The bottom-line improvement over BIFI is not significant. I do appreciate that it is hard to improve every basis point beyond 90.5% obtained by BIFI. I also understand that this is obtained without a Breaker model.

- The claim that LLMs tend to make more global changes seems plausible, but you can probably control for that with prompt engineering. So the comparison with LLMs ability to fix these errors is not giving LLMs the full ability to address the problem as defined.

**Questions:**

- You write "A key contribution of our work is to significantly simplify the process of training a syntax fixer of (slightly) higher quality than prior work (viz., BIFI)." - is this process a bottleneck for applying the technique? What is the cost/barrier for applying BIFI that is significantly improved by FILI?

- Do you have any hypothesis on why you did not see further improvement beyond two rounds?

- Can you try experiments with LLMs when providing them with instructions to only make local modifications? How would that look?

- page 7: should be "FILI cannot solve 1263, while BIFI cannot solve 1428"?

---

> ### Author Response · Authors · 2023-11-21
>
> **Hypothesis for no improvement in test-set performance after 2 rounds:**
>
> We conducted a manual inspection of examples that both BIFI and FILI cannot fix. We randomly sampled 15 incorrect programs from the test set and then checked whether these programs could be corrected or not. Through this manual inspection, we discovered that only 6 out of these 15 could be repaired by fixing the syntax errors. In the remaining 9 instances, we observed that they are not solely syntax repair problems. For instance, some of these programs are incomplete and require generating additional program statements for correction. Similarly, some of these programs involve deleting specific program statements, which neither FILI nor BIFI is trained to perform. Finally, we noted that some programs were written in other languages such as C++ and Java. This observation indicates that the upper limit for success on this test set is not 100%, and a careful analysis is needed to determine how many programs are actually fixable. Consequently, a 1% improvement over BIFI might indeed be reasonable within this context.
>
> **cost/barrier for applying BIFI that is significantly improved by FILI?**
>
> With our ablation study, we demonstrate that curriculum learning is helpful in generating more parsable programs.  From our experiments, we observe that curriculum learning helps improve the performance of BIFI also. Training the full BIFI model with curriculum would require more careful analysis, owing to the added complexity introduced by the breaker model. Additionally, it’s important to note that there exists several options to consider when deciding whether to apply curriculum learning to the breaker or the fixer, as well as how to effectively combine the data obtained from the beam and the breaker.

---

### Official Review · Reviewer_9dQP · 2023-11-02

**Soundness:** 3 good
**Presentation:** 3 good
**Contribution:** 2 fair
**Rating:** 3
**Confidence:** 5

**Summary:**

This paper proposes FILI (Fix-It-Learn-It), which simplifies a previous work -- BIFI (Break-It-Fix-It), an unsupervised learning approach for fixing syntax errors in programs. BIFI requires two trained models (i.e., a breaker and a fixer), while FILI requires only one fixer model. The observation is that the fixer model is not perfect and thus generates correct fixes as well as incorrect fixes. In the latter case, the fixer model itself can be viewed as a breaker model. Instead of training a separate breaker model, which can be expensive, one fixer model can be used to generate both good programs and bad programs. The evaluation on the same dataset shows that FILI slightly outperforms BIFI.

**Strengths:**

- Compared to the previous work BIFI, LIFI is simple, more efficient, and achieves (slightly) better performance.
- Extensive evaluations and comparisons with BIFI are performed on the original dataset.

**Weaknesses:**

- The idea of using a fixer as a breaker is fairly incremental, and the improvement of performance is quite minor. Given that BIFI already achieves 95.5% accuracy over the chosen dataset, further improving it to 96.1% adds little value.
- Curriculum learning only makes very small differences and thus seems not an essential part of the LIFI.

**Questions:**

In Table 2, two accuracy scores (the last two columns) are reported. Can you elaborate on the key difference? Why is there a sharp drop for all approaches, especially GPT-3.5-turbo?

Is there any particular reason that a breaker model is more difficult to train? Page 6 mentions that training a fixer for two rounds takes around 20 hours, while a breaker model requires 2 days.

The dataset collected by BIFI seems pretty much saturated. Have the authors considered a different dataset? (A comment rather than a question).

There is a minor typo at the bottom of page 7, "BIFI cannot solve 1263, while BIFI cannot solve 1428". BIFI was mentioned twice, one of which should be LIFI.

---

> ### Author Response · Authors · 2023-11-21
>
> **In Table 2, two accuracy scores (the last two columns) are reported. Can you elaborate on the key difference? Why is there a sharp drop for all approaches, especially GPT-3.5-turbo?**
>
> The first column corresponds to the parse rate without considering the edit distance between the incorrect source program and the predicted program from the models. The models can introduce an arbitrary number of changes to the programs, which is undesirable. An ideal syntax fixer should solely address syntax errors in the programs, leaving the other parts unchanged. The second column corresponds to the parse rate with the edit distance, wherein the predicted program is considered correct only if it is both parsable and below a certain edit-distance threshold. Specifically, we set the edit-distance threshold to 4 to align with BIFI's evaluation criteria.
>
> The significant difference in language model performance between the two columns is because it is challenging to constrain the model's output using natural language prompts to only rectify syntax errors. In addition to addressing syntax errors, these models end up making numerous changes to other parts of the code. Notably, they introduce stylistic changes and alter variable declarations by combining them, among other things. A detailed explanation, along with examples, is provided in Appendix A.6 regarding the outputs from these pre-trained language models.

---

### Author Response · Authors · 2023-11-21

We thank the reviewers for their helpful comments and suggestions. We will incorporate all suggestions and clarify the confusion in our next version. Below, we address the concerns that the reviewers raised.

**Should be "FILI cannot solve 1263, while BIFI cannot solve 1428"?**

Thank you for pointing this out, it was a small typo and indeed it should be FILI cannot solve 1263, while BIFI cannot solve 1428.

**LLMs experiment: instruction for making only local changes**

We prompt the model to make only local changes to the program and not make any other changes. We use the following two prompts and evaluate both Bard and GPT-3.5 on the test-set:

Prompt-1:

'''

The following python program has syntax errors. Fix all the syntax errors to make the program parsable. Do not make any other changes to the program. Just fix the syntax errors.
<Incorrect Program>:
```
{{incorrect}}
```
<Correct Program>:

'''

Prompt-2:

'''

The following python program has syntax errors. Fix all the syntax errors to make the program parsable. Do not make any other changes to the program. Keep the edit distance between the original program and the fixed program as small as possible.
<Incorrect Program>:
```
{{incorrect}}
```
<Correct Program>:

'''
-------------------------------------------------------------------------------
| Model    | Prompt | Edit = 4 | Edit = 6 | Edit = 8 | Edit = 10 | Edit = inf |
|----------|--------|----------|----------|----------|-----------|------------|
| Bard 1    | 1      | 55.15%   | 68.31%   | 73.55%   | 75.8%     | 78.09%     |
| GPT-3.5  | 1      | 46.47%   | 59.72%   | 68.6%    | 76.8%     | 98.5%      |
| Bard        |2    | 55.64% | 69.8%    | 75.92%   | 78.58%   | 81.34%    |            |
| GPT-3.5  | 2      | 46.76%   | 60.4%    | 69.2%    | 77.6%     | 98.69%     |


Despite instructing the models not to modify other sections of the program through prompts, the edit distance between the predicted program and the incorrect source program exceeds the threshold used by BIFI (and FILI) for evaluation. Consequently, although LLMs are good in fixing syntax errors, they introduce changes to other parts of the program.


**Accuracy gains are small**

We agree with the reviewers that FILI does not significantly outperform BIFI in terms of accuracy. With FILI our major contribution is significantly simplifying the training of fixer in an unsupervised setting while still achieving slightly better accuracies compared to BIFI, the previous state-of-the-art approach for training syntax repair models. For training FILI, it took ~20 hours to train with one TPU (v3-8) and for training the full BIFI model with the breaker training it takes ~72 hours. BIFI involves training the fixer and the breaker model which is expensive. Moreover, the BIFI framework also involves making an inference on ~3 million synthetic data examples using the trained breaker model to augment data for the fixer training which can be very expensive. In contrast, FILI uses the beam predictions to augment data which is significantly cheaper to perform than doing an inference on the synthetic data. Overall FILI’s approach is more resource efficient while maintaining the similar accuracies as the previous models. The efficiency in training is particularly important in light of the fact that obtaining compute is challenging.

**Cost/difficulty of training breaker model in BIFI:**

BIFI trains the breaker model with significantly more data than the fixer model. The training of the breaker model involves using the <good programs, bad programs> pair. To generate the training data, BIFI applies the breaker model to the good programs, producing the corresponding incorrect programs. The overall training of the breaker model takes more time compared to the fixer because there are 3 million good programs (syntactically correct) on which the model needs to perform inference to generate additional data. This inference must be carried out for each training iteration of BIFI, as the breaker model is updated after every iteration, resulting in an expensive training procedure. On the other hand, the fixer's additional training data is generated by performing inference on only ~20k examples, leading to a more cost-effective training procedure than the breaker model.

---

### Meta-Review · Area_Chair_4HZK · 2023-12-06

**Metareview:**

The authors unanimously raised a couple of concerns, including (1) relatively incremental improvements in some cases, and (2) a relatively reasonable question about whether the tasks considered in the paper are solvable by well prompted LLMs. While the authors point that LLMs take significant additional resources to train is well taken, ultimately these models do exist. The authors point out that the LLMs they tested made code edits in unintended areas, which would be an interesting limitation. However, I'd like to see this limitation studied more systematically with experimental setup given. The authors mention that the instructed the LLMs to not do this, but it's not clear if this problem couldn't be solved, for example, by simply constructing a better prompt. Ultimately, I think it's fair to want these questions answered, and the paper would be much stronger for it.

**Justification For Why Not Higher Score:**

The authors could include a more complete and detailed comparison to off-the-shelf LLMs, which would help justify their approach significantly.

**Justification For Why Not Lower Score:**

N/A

---

### Decision · Program_Chairs · 2024-01-16

Reject